# Breeding Maize Hybrids with Improved Drought Tolerance Using Genetic Transformation

**DOI:** 10.3390/ijms251910630

**Published:** 2024-10-02

**Authors:** Zhaoxia Li, Juren Zhang, Xiyun Song

**Affiliations:** 1Agronomy College, Qingdao Agricultural University, Qingdao 266109, China; songxiyun2020@163.com; 2Key Laboratory of Plant Development and Environment Adaptation Biology, Ministry of Education, School of Life Sciences, Shandong University, Qingdao 266237, China; jrzhang@sdu.edu.cn

**Keywords:** *Zea mays*, *betA*, drought tolerance, glycine betaine, hybrid

## Abstract

Drought is considered the main agricultural menace, limiting the successful realization of land potential, and thereby reducing crop productivity worldwide. Therefore, breeding maize hybrids with improved drought tolerance via genetic manipulation is necessary. Herein, the multiple bud clumps of elite inbred maize lines, DH4866, Qi319, Y478 and DH9938, widely used in China, were transformed with the *Escherichia coli betA* gene encoding choline dehydrogenase (EC 1.1.99.1), a key enzyme in the biosynthesis of glycine betaine from choline, using *Agrobacterium* to generate *betA* transgenic lines. After 3–4 consecutive generations of self-pollination in these transgenic plants, progenies with a uniform appearance, excellent drought tolerance, and useful agricultural traits were obtained. We evaluated the drought tolerance of T4 progenies derived from these transgenic plants in the field under reduced irrigation. We found that a few lines exhibited much higher drought tolerance than the non-transformed control plants. Transgenic plants accumulated higher levels of glycine betaine and were relatively more tolerant to drought stress than the controls at both the germination and early seedling stages. The grain yield of the transgenic plants was significantly higher than that of the control plants after drought treatment. Drought-tolerant inbred lines were mated and crossed to create hybrids, and the drought tolerance of these transgenic hybrids was found to be enhanced under field conditions compared with those of the non-transgenic (control) plants and two other commercial hybrids in China. High yield and drought tolerance were achieved concurrently. These transgenic inbred lines and hybrids were useful in marginal and submarginal lands in semiarid and arid regions. The *betA* transgene can improve the viability of crops grown in soils with sufficient or insufficient water.

## 1. Introduction

Maize is a versatile muti-purpose crop that is primarily used as a food and feed crop globally but also as a raw material of industrial products including starch, sweeteners, oil, industrial alcohol and fuel ethanol. Global maize production has surged in the past few decades, and it is already the leading cereal in terms of production volume and is set to become the most widely grown and traded crop in the coming decade. Maize production is critical for guaranteeing world food security and people’s life quality. Abiotic stresses, such as drought, salinity, and extreme temperatures, are major limitations in crop productivity [1]. According to the report of IPCC, the global average surface temperature has increased by around 1.1 °C compared with the average between 1850–1900 [2]. Due to global warming, extreme climate events are becoming more and more frequent. Drought frequency and severity are projected to increase as a consequence of global warming. Moreover, drought can amplify heat waves, and compound drought and heat or drought and salinity events cause more severe impacts than alone. Extreme climate events such as drought, heat, salinity, and their combinations are major menaces in rain-fed agricultural regions and severely impact crop yields worldwide [3,4]. Rainfall varies annually, and irrigation is difficult in some maize-growing areas in China and Africa, which could threaten decent maize yields. Depending on the intensity or duration of drought stress and crop stage, the maize yield losses vary from 30 to 90%, severely affecting the flowering and grain-filling stages [5]. Boost crop resistance is the most effective way to reduce climate change in agriculture; therefore, breeding maize cultivars that are drought-tolerant is necessary and urgent.

Drought stress results in complex alterations in physiological and biochemical processes in plants. Osmotic regulation is a crucial strategy for plants to reduce their osmotic potential and maintain or minimize damage to the biochemical, physiological, and morphological processes related to cell growth, stomatal opening, and photosynthesis during environmental stress [6,7]. Plants can osmoregulate (maintain osmobalance) to various degrees to adapt to changes in the osmotic pressure of the environment. There is a large variation in the responses of plants to drought stress, such as synthesizing osmoprotectants, enhancing the anti-oxygenation defense system, modifying photosynthesis, enhancing water uptake, and maintaining osmotic pressure balance [7,8]. Multiple osmoprotectants, including proline, glycine betaine (GB), γ-aminobutyric acid, and sugars, have been identified as effective components for stress tolerance in different plant systems [7,9]. GB [(CH_3_)_3_N^+^CH_2_COO^−^]) is among the most extensively studied compatible solutes [10,11,12,13,14].

GB acts as an osmoregulator, and its accumulation in cells under salt stress relaxes ionic toxicity and maintains normal osmotic pressure. In addition, GB stabilizes the structures and activities of enzymes and protein complexes and maintains the integrity of membranes against the damaging effects of drought stress [15,16,17]. Genes associated with GB synthesis in higher plants and microbes have been transferred into plants that do not accumulate GB, e.g., tobacco (*Nicotiana tabacum*) [18,19,20], *Arabidopsis thaliana* [13,21,22,23], *Brassica napus* [24], and rice (*Oryza sativa*) [25,26]. The metabolic engineering of GB biosynthesis in these transgenic plants improves their tolerance to salt, drought, and extreme temperature stress [14,27]. However, most of these studies focused on model plants.

GB levels vary among different maize varieties [28]. A positive correlation was found between endogenous GB levels and the degree of salt tolerance in maize [29]. In our lab, we expressed *betA* from *Escherichia coli* in maize, and the transgenic maize acquired improved chilling or drought tolerance because of the enhanced accumulation of GB [30,31]. In *E. coli*, GB is synthesized from choline via a two-step pathway wherein choline dehydrogenase (CDH, EC 1.1.99.1), encoded by *betA* [32,33], catalyzes the oxidation of choline into betaine aldehyde. The second step is catalyzed by betaine aldehyde dehydrogenase (EC 1.2.1.8) encoded by *betB* [32,33]. CDH can also catalyze the second step, i.e., the oxidation of betaine aldehyde to GB [34]. 

Maize is an important food and fodder crop. Several transgenes have been transferred to maize to improve the traits of cultivars. In recent years, several studies have identified thousands of novel loci and candidate genes for drought-stress tolerance in maize [1,5,35,36,37]; however, most of these studies have been conducted under controlled laboratory or greenhouse conditions and using maize inbred lines, not in field conditions or with maize hybrids [1,36,38,39]. Herein, we introduced *betA* of *E. coli*, which codes for CDH, into four maize elite inbred lines DH4866, Qi319, Ye478, and DH9938, widely used as parent lines in maize hybrid breeding in northern China, using *Agrobacterium tumefaciens*-mediated transformation. The drought tolerance test was conducted on the transgenic lines wherein the *betA* transgene was integrated at a single locus in the genome and exhibited high *betA* expression levels. Transgenic lines from DH4866, Qi319, Ye478, and DH9938 were used to evaluate dehydration tolerance at the seed germination and three-leaf stages under laboratory conditions and the flowering stage under field conditions. Moreover, the plants derived from the inbred lines were evaluated for their drought tolerance, GB content, agronomic traits, and physiological changes under different environmental conditions. Lines with excellent agronomic traits and drought tolerance were used for crossbreeding. The drought tolerance and agronomic traits of hybrids derived from these transgenic lines were found to be enhanced under field conditions.

## 2. Results

### 2.1. Identification of the Transgenic Plants

The elite inbred lines of maize, DH4866, Qi319, Ye478, and DH9938 were used as transgenic donor lines. The transfer DNA region of plasmid pCU-*betA-als* contains *betA* encoding CDH and a selectable marker gene *mutant acetolactate synthase* (*als*) encoding mutant acetolactate synthase (Figure 1a). The presence of the transgene in the regenerated plants and their progenies was validated by polymerase chain reaction (PCR) analysis and further confirmed by Southern blot hybridization. A 1.1 kb *als* fragment, with the same size as that of pCU-*betA*-*als*, could be amplified from the DNA of candidate transgenic plants in the PCR analysis, indicating the existence of the transgene in the plants; however, no amplification of DNA was detected in the DNA samples from non-transgenic (NT) plants (Figure 1b). Southern blot hybridization with DNA from PCR-positive transgenic plants was performed (Figure 1c). Of 1809 regenerated plants, 651 were PCR-positive, and the numbers of PCR-positive plants from the inbred lines DH4866, DH9938, Ye478, and Qi319 were 264, 125, 132, and 130, respectively. Of 260 randomly selected PCR-positive plants, 233 were confirmed to be transgenic based on the results of the Southern hybridization analysis. Northern blot analysis of transgenic plants showed varying levels of *betA* expression in transgenic plants (Figure 1d). Transgenic lines with high *betA* expression levels were used for further analysis.

A total of 40 and 30 independent T_1_ transgenic lines derived from the T_0_ transgenic plants of DH4866 and Qi319, respectively, were randomly selected for further genetic analysis. Plants from five of these independent transgenic lines showed abnormal morphology and were excluded from the analysis at the pollination stage. However, plants derived from the remaining 65 independent transgenic lines grew normally, except for some differences in the growth of these transgenic plants and the wild-type NT plants. In a few T_1_ or T_2_ transgenic lines, the transgene was found to be segregating in Mendelian ratios (Appendix A). In the T_1_ generation, DH4866-003 and Qi319-007 showed a 3:1 segregation ratio, indicating that the transgene was integrated at a single locus in their genome, and DH4866-005 and Qi319-103 showed a 15:1 segregation ratio, indicating that the transgene was integrated at two independent loci in their genome. Based on the results of the PCR assays and agronomic traits, some PCR-positive T_1_ plants were selected for self-pollination to produce T_2_ lines. In the T_2_ lines derived from T_1_ plants, the transgene continued to segregate in Mendelian ratios, indicating that the transgene was stably transferred from T_1_ to T_2_ plants and that some transgenic plants were homozygous for the transgene in the T_1_ generation. For example, lines DH4866-0511, DH4866-0512, and DH4866-0513 were derived from DH4866-051 and showed segregation ratios of approximately 3:1 or 30:0 segregation ratios for the transgenic *betA* gene.

### 2.2. Screening of Inbred Lines for Drought Tolerance

The transgenic plants were selected and self-pollinated to produce seeds. After 3–4 generations, the plants in one line showed unified characteristics. Homologous lines with useful agronomic traits were selected to examine the physiological changes in these lines under both normal and drought-stress conditions to determine the stability and function of the transgene (Figure 2). Transgenic lines derived from DH4866, Qi319, Ye478, and DH9938 were used to evaluate dehydration tolerance. When seeds (T_4_) were germinated in Murashige and Skoog (MS) inorganic salt solution, the germination capacity of the transgenic seeds was similar to that of the seeds derived from the NT plants. However, when seeds were germinated in MS inorganic salt solution with 20% (*w*/*v*) polyethylene glycol (PEG) 6000, the germination was delayed for nearly 1.5 days; moreover, the time to 50% germination continued to be 0.5 days faster with the final germination percentage being 2–5% higher in the seeds of certain transgenic lines than that in the seeds derived from the NT plants (Figure 2a). The primary roots and shoots of the *bet A* transgenic lines were significantly longer than those of NT when germinated in MS inorganic salt solution with PEG 6000 (Figure 2a,b).

When the three-leaf stage seedlings cultured in MS salt solution were transferred to fresh MS salt solution supplemented with 14% (*w*/*v*) PEG 6000 for an additional 7 days of culture, they showed retarded growth and dehydration with droopy leaves to distinct degrees (Figure 2c). The leaves of the control seedlings drooped after treatment for 2 d, whereas the seedlings of some transgenic lines showed drooping leaves after 6 d of dehydration. For example, the progenies of DH4866-003111, DH4866-042311, DH4866-051211, DH4866-012311, DH4866-014811, and Qi319-032111 displayed higher drought tolerance than that of the NT lines and continued growing to the five-leaf stage under this treatment. An examination of the survival frequencies of the transgenic lines and NT control plants of DH4866 after 12 days of osmotic stress revealed survival frequencies of 50, 71, 56, and 69% in DH4866-003111, DH4866-042311, DH4866-051211, and DH4866-012311, respectively, indicating improved drought tolerance in the seedlings; however, the survival frequency of the NT plants was only 26%, and that of DH4866-109111 was 32%. Transgenic lines showing improved drought tolerance, both in seed germination speed and seedling dehydration tests, were grown in wooden troughs and in the field to examine the physiological changes in these lines under different environments and their abiotic tolerance.

### 2.3. GB Content

Five transgenic lines derived from DH4866 were selected to determine their GB concentrations and the physiological changes in these lines under different environmental conditions. The GB content in the seeds of NT plants was approximately 2.12 μmol/g dry weight (DW), whereas that in the seeds of transgenic lines DH4866-003111, DH4866-012311, DH4866-042311, and DH4866-005211 was 6.80–8.32 μmol/g DW, much higher than that in the seeds of NT plants, and only the seeds of the DH4866-109111 line had a GB content of 3.70 μmol/g DW, which was slightly higher than that in the seeds of NT plants (Figure 3a and Appendix A).

The three-leaf stage seedlings derived from the transgenic lines and NT plants were transferred to MS salt solution containing 14% (*w*/*v*) PEG 6000 and cultured for 5 days. Before the osmotic treatment in the solution, GB concentrations in leaves were 1.41 μmol/g fresh weight (FW) in NT plants, 2.59 in DH4866-109111, and 5.12, 5.42, 5.99, and 5.84 μmol/g FW in DH4866-003111, DH4866-012311, DH4866-042311, and DH4866-051211, respectively. After being subjected to osmotic stress for 5 days, the concentrations of GB significantly increased in the leaves of all lines but continued to be significantly higher in the transgenic lines DH4866-003111, DH4866-012311, DH4866-042311, and DH4866-051211 than that in the NT plants (2.23 μmol/g FW) (Figure 3b and Appendix A). Choline concentrations in leaves and seeds showed no significant differences between the transgenic plants and the NT control plants, although there was a 30% increase in leaf choline concentrations after osmotic stress.

### 2.4. Photosystem II Activity and Net Photosynthesis Rate

Maize seedlings at the three-leaf stage were transferred to MS salt solution containing 14% (*w*/*v*) PEG 6000, and the activity of photosystem II (*Fv*/*Fm*) was measured per day. The results showed no difference in *Fv*/*Fm* between transgenic lines and NT before osmotic stress; however, during osmotic treatment, the decrease in *Fv*/*Fm* in transgenic lines was less than in the control NT plants. After 5 days of osmotic stress, the *Fv*/*Fm* of the NT plants was reduced to 70% of the initial activity, whereas it was 80–85% in the transgenic lines (Figure 4).

Before osmotic stress, the net photosynthesis rate on the third leaf of each three-leaf stage seedling was approximately 16.5 μmol CO_2_/m^2^/s in both the transgenic and control NT plants. However, after 5 days of treatment, the net photosynthesis rate of the control NT plants decreased to 36% of the initial rate, whereas it was 44–56% reduced in the transgenic lines DH4866-003111, DH4866-012311, DH4866-042311, and DH4866-051211 (Figure 4). This result implies that the photosystem of the transgenic plants is relatively more stable than that of the control NT plants under the conditions of osmotic stress. 

### 2.5. Damage to Leaf Cell Membranes and Relative Water Content

We determined the damage to leaf cell membranes in the three-leaf stage seedlings from the transgenic and NT lines subjected to the 14% (*w*/*v*) PEG 6000 treatment based on electrolyte leakage. When treated with PEG 6000, leaf cell membrane damage increased gradually in all lines; however, the damage to transgenic lines was less than that of NT. For example, after 5 days of osmotic treatment, the damage to the leaf cell membranes in the NT plants was 60%, whereas that in the transgenic line DH4866-042311 was only 48% (Figure 5). Relative water content (RWC) is considered as an indicator of plant status. Before being subjected to stress, no significant differences in RWC were found between the transgenic and NT plants; however, under osmotic stress, a higher rate of water loss was observed in the NT plants, indicating higher RWC in the transgenic plants than in NT plants (Figure 5b).

### 2.6. Drought Treatment of Maize Transgenic Lines at the Pre-Flowering and Heading Stages

T_4_ transgenic plants grown in wooden troughs or flowerpots were irrigated sufficiently every 2 days until the 10-leaf stage. Subsequently, the RWC of soil was maintained at approximately 15% for 3 weeks by reducing the quantity of irrigation for a drought-stress treatment at the pre-flowering stage, following which the plants were irrigated sufficiently (Figure 2d). The net photosynthetic rate and RWC of the first fully expanded leaf (from the top) were evaluated (Figure 5c,d). As shown in Figure 2d, the plants derived from the transgenic lines wilted relatively less than the NT plants, which is consistent with the results of the RWC analysis—the RWC of the plants derived from the transgenic lines was 104.1–109% of that in the NT plants. After 5 days of treatment, the net photosynthesis rate of the control NT plants decreased; however, that of the transgenic lines DH4866-003111, DH4866-012311, DH4866-042311, and DH4866-051211, particularly of the lines DH4866-012311 and DH4866-042311, was significantly higher (141.9–198%) than that of the NT plants.

For drought treatment at the heading stage, the transgenic and NT plants at the 14-leaf stage grown in a rain exclusion shelter were subjected to dehydration stress for 2 weeks, during which the RWC of soil at a depth of 40 cm was maintained at 15–18%; subsequently, the plants were watered sufficiently until maturation (Figure 2e–g). The treated plants showed the distorted development of tassels and ears, the increased interval of anthesis and earing period (from 1 to 2 days under normal conditions to 4 to 8 days under the drought treatment), and a much lower yield of kernels than that in the plants grown in normal environments. The tassels were short and had fewer branches, with the number of pollen grains reduced significantly than in the plants grown in normal environments. The ears were small, had fewer kernels, and the kernel weight was low (Figure 2f,g and Appendix A). The transgenic plants showed a much better appearance of tassels and ears and kernel yield than that of NT, and the interval between anthesis and earing period was usually decreased by 3–4 days compared with that in the control plants. The per plot yields of the DH4866-00311, DH4866-012311, DH4866-042311, and DH4866-051211 lines showed an increase of 11.66, 22.96, 26.17, and 22.81%, respectively, compared with that in the NT lines. Based on the above physiological and agronomic parameters, the introduction of the *betA* transgene into elite maize inbred lines improved the drought tolerance of the transgenic inbred lines.

### 2.7. Breeding of Drought-Tolerant Hybrids

Based on the results of the drought tolerance analysis, a few transgenic lines were chosen to further examine their combining ability. Transgenic lines derived from DH4866 with adequate drought tolerance were used as female parents in a cross with the inbred line 196 as a pollen donor to create transgenic hybrids. In the normal field, out of 48 transgenic hybrids, 28 did not show a significant difference in yields compared with those of the NT hybrid Denghai 1^#^ (DH4866 × 196); 10 showed lower yields, whereas another 10 showed higher yields than those of Denghai 1^#^, an elite commercial hybrid widely used in China. Therefore, the specific combination ability of most transgenic lines did not distinctly change in the self-pollinating generations.

When the hybrids in the heading period were stressed by controlled irrigation, they showed droopy leaves and retarded development. However, the transgenic hybrids showed lighter wilting at noon and were restored relatively quicker than the NT plants after normal watering. Under 15 days of drought stress, the transgenic hybrids produced higher yields than the NT hybrids (Figure 6a and Appendix A). Among these, all of DH4866-012311 × 196, DH4866-042111 × 196, and DH4866-042311 × 196 gave 20% higher yields than those of Denghai 1^#^ (DH4866 × 196) (Appendix A).

To evaluate the commercial importance of the transgenic hybrids, we cultivated transgenic hybrids with improved drought tolerance along with two commercial hybrids available in China—Zheng958, a drought-resistant cultivar, and Denghai 9^#^, a new high-yield cultivar—under both normal and no irrigation conditions at the heading stage. As shown in Figure 6c,d and Appendix A, the yields of the transgenic hybrids were significantly higher than those of the commercial hybrids under normal conditions. Moreover, under the no irrigation condition at the heading stage, the yields of the transgenic hybrids were much higher (25.10–33.13%) than those of Denghai 9^#^ and significantly higher (8.81–17.46%) than those of Zheng 958 (Figure 6d and Appendix A).

Based on the above results, we conclude that the introduction of *E. coli betA* into elite maize inbred lines significantly improved the drought tolerance of the transgenic inbred lines. Several drought-tolerant hybrids with high yields, useful in marginal and submarginal lands in semiarid and arid regions, were bred from the progenies of the transgenic inbred lines.

## 3. Discussion

Maize is among the most important food and feed crops and is susceptible to drought during its growth and development, particularly at the flowering stage [40]. Drought is a major stress in maize production and is usually accompanied by other stresses, such as heat or salinity. With the current trends and predictions of future greenhouse gas emissions, global water source shortages, and an increasing world population, plant scientists and breeders are facing a time-sensitive challenge in breeding drought-tolerant maize and improving crop yields [41,42]. 

Genetic engineering, i.e., the manipulation of genetic diversity through transformation and genome editing, involves the specific handling of a few genes as compared to classical breeding, wherein thousands of genes are rearranged [41,42]. As genes can be obtained from other species or even synthesized in the laboratory, scientists are not limited by genetic variation within a crop species. With the help of omics approaches and genome-wide association studies, thousands of novel loci and candidate genes for drought-stress tolerance have been identified in maize; however, most of these studies were conducted in controlled laboratories or greenhouses using maize inbred lines, and not using maize hybrids and field conditions.

In this study, *betA*, coding for CDH in *E. coli*, was introduced into four elite inbred maize lines, DH4866, Qi319, Ye478, and DH9938, widely used as parent lines in maize hybrid breeding in northern China. Herbicide screening, PCR, and Southern and Western blotting were used to select transgenic lines with high *betA* expression levels. The drought tolerance test was conducted on the transgenic lines wherein the *betA* transgene was integrated at a single locus in the genome and exhibited high *betA* expression levels. Moreover, dehydration tolerance at both the seed germination and three-leaf stages was evaluated using the transgenic lines derived from DH4866, Qi319, Ye478, and DH9938. When seeds were germinated in MS solution with 20% (*w*/*v*) PEG 6000, germination was delayed for nearly 1.5 days. In addition, the germination of the transgenic lines with high GB contents was much better than that of the NT plants and the transgenic line with low GB content. When grown on MS salt solution supplemented with 14% (*w*/*v*) PEG 6000, growth and leaf color were healthier in the transgenic plants than in the NT plants. 

The GB content, photosystem II activity, net photosynthesis rate, cell membrane damage, and RWC were monitored when plants from different lines were transferred into the MS salt solution containing PEG 6000. There was no statistical difference between transgenic lines and NT in *Fv*/*Fm*, net photosynthesis rate, cell membrane damage, and RWC before osmotic stress, although the GB content in the leaves of transgenic lines was significantly higher than that in NT, except for line DH4866-109111. When subjected to osmotic stress, the photosystem II activity and net photosynthesis rate gradually declined, whereas cell membrane damage gradually increased in all lines. However, the declines in the photosystem II activity and net photosynthesis rate in the transgenic lines were slower than those in the NT plants. In contrast, the photosystem II activity and net photosynthesis rate in the transgenic lines were higher than those in the NT plants at the same time points after the removal of osmotic stress. An opposite trend in cell membrane damage was observed between the transgenic lines and the NT plants. The GB content correlated positively with photosystem II activity and net photosynthesis rate. These results suggest that *betA*-transgenic plants enhance drought tolerance by enhancing photosynthesis and cell membrane integrity because the transgene confers protection against drought. 

Maize plants at the heading stage were subjected to drought treatment in the rain exclusion shelter to test whether enhanced GB levels help the plants cope with drought stress at the reproductive and final yield stages. The agronomic traits of the transgenic lines, including the interval between anthesis and earing period, tassel branches, kernel number, yield per plant, and yield per plot, were relatively less affected by the drought-stress treatment. Thus, the transgenic lines showed improved agronomic parameters than those of the NT plants. The kernel weights and yields in the *bet-A* transgenic lines were 20 and 40% higher, respectively, than those in the NT plants. 

Transgenic lines with adequate drought tolerance were used as female parents to create transgenic hybrids, and the performance of the transgenic hybrids was evaluated in the field using an NT hybrid and two other commercial hybrids used in China. The crop yields of the transgenic hybrids were 20% higher than those of the NT plants and significantly higher than those of the commercial hybrids under both normal and stress conditions. 

In recent years, the role of regulatory proteins in drought tolerance has received increasing attention because they trigger a cascade of genes that act together to enhance stress tolerance. However, the altered expression of downstream genes in all organs and developmental stages can cause abnormalities in plants grown under normal conditions. The accumulation of compatible solutes or osmolytes is a common strategy many organisms adopt to cope with environmental stresses such as drought and salt stress. However, the constitutive accumulation of compatible solutes like polyamines, proline and trehalose resulted in abnormal development and growth [43,44]. For maize breeding, both the stress tolerance and high yield need to be considered. The balance between stress response and plant growth is crucial for engineering crops with improved stability in the field. GB is a quaternary amine with a zwitterionic nature, and its natural accumulation in cells is known to protect organisms against abiotic stresses via osmoregulation or osmoprotection [45,46,47]. Moreover, the stress-related transcriptome was affected by the limited accumulation of GB and may also contribute to stress tolerance. However, maize did not accumulate a significant amount of GB naturally, which could be due to the alternative splicing of the endogenous betaine aldehyde dehydrogenase coding gene. It could be considered a potential target for the metabolic engineering of GB biosynthesis by introducing the GB biosynthetic pathway. In this study, the introduction of *betA* in four inbred lines improved drought-stress tolerance without affecting yield potential. The drought tolerance of the transgenic hybrids was enhanced under field conditions compared with that of NT and the other two commercial hybrids used in China. The yield increased under both normal and drought-stress conditions, indicating that high yield and drought tolerance were achieved concurrently. Transgenic inbred lines and hybrids are useful in marginal and submarginal lands in semiarid and arid regions. The *betA* transgene can improve the viability of crops grown in soils with sufficient or insufficient water.

## 4. Materials and Methods

### 4.1. Plant Materials

Four elite inbred lines of maize—DH4866, Qi319, Ye478, and DH9938—widely grown in northern China, were used for all experiments. Surface-sterilized seeds were germinated on a modified MS medium in the dark. When aseptic seedlings were 3–5 cm long, the coleoptile and young leaves were peeled off, and the apices were cultured in the dark on the modified MS medium supplemented with multiple combinations of 6-benzylaminopurine and 2,4-dichlorophenoxyacetic acid to induce multiple bud clumps, as previously described [48]. Transgenic plants were obtained using *A. tumefaciens-*mediated transformation [48]. The *A. tumefaciens* strain LBA4404 harboring the plasmid pCU-*betA-als*, which contains *betA* encoding CDH and the selectable marker gene *als* encoding mutant acetolactate synthase, was used (Figure 1). Transgenic lines at T_4_ generation, which showed physical traits similar to those of the NT plants under normal conditions, were used in the experiments.

### 4.2. Molecular Identification of the Transgenic Lines

Genomic DNA was extracted from the leaves of the transformed plants at the six-leaf stage using the cetyltrimethylammonium bromide protocol. The sequences of PCR primers for *als* (safe for humans and other mammals) derived from a mutant *A. thaliana* line were as follows: P1, 5′-ACAGGACAAGTCTCTGGTCG-3′; P2, 5′-GGGTTAGCAACAGACGCT-3′. These primers could not amplify the endogenous *als* gene in maize. The reaction conditions were identical to those reported previously [49]. The sequences of primers for *betA* were as follows: P1, 5′-CTACCCGTCTGACTGAAGATC-3′; P2, 5′-CCCATTTGCCACAAAATA TCC-3′. PCR was performed with 35 annealing cycles at 58.5 °C to amplify a 1.6 kb sequence.

For Southern blot analysis, maize genomic DNA digested with *Kpn*I was hybridized to a digoxigenin (DIG)-labeled DNA probe for *betA* transgene. Hybridization protocols were implemented following the manufacturer’s instructions for the DIG High Prime Labeling and Detection Starter Kit (Roche), as reported previously [49]. The probe was a DIG-labeled *betA* sequence cut from pCU-*betA-als*.

Total RNA (30 mg) was separated on a 1.2% agarose formaldehyde gel and transferred to a nylon membrane. Prehybridization and hybridization were performed as described in Molecular Cloning: A Laboratory Manual (3rd edition) with [a-^32^P]dCTP-labeled *betA* as probe in Church buffer at 65 °C.

### 4.3. Osmotic Treatment of Seeds during Germination and Seedlings at the Three-Leaf Stage

Surface-sterilized seeds derived from the T_4_ homogeneous transgenic plants and NT plants were soaked by sterilized water and then cultured in Petri dishes containing two layers of filter paper moistened with MS salt solution plus 20% (*w*/*v*) PEG 6000 at 25 °C in the dark. Germination percentage was recorded daily. Seeds germinated without PEG 6000 were used as controls.

Germinated seedlings were cultured in the MS inorganic salt solution until the three-leaf stage. Subsequently, some of these seedlings were transferred to MS inorganic salt solution containing 14% (*w*/*v*) PEG 6000 for another 7 days to analyze seedling morphology and GB content and measure physiological parameters. All cultures were aerated continuously at 25 °C with an average photon flux density (PFD) of 300 μmol/m^2^/s and a light/dark cycle of 14/10 h per day.

### 4.4. Determination of GB Content in Maize Seeds and Leaves

Dry maize seeds were ground to powder and incubated with redistilled water at 4 °C for 24 h. Maize leaves were frozen in liquid nitrogen and thawed to exude sap using a glass injector. The seed powder solution and leaf sap were centrifuged for 15 min at 12,000 rpm at 4 °C, and 50 μL supernatant of each sample was dried in a stream of N_2_ gas. The dried fractions were dissolved in D_2_O containing 1.0 mM 3-(trimethylsilyl)-propane-sulfonic acid sodium salt (DSS), which acted as an internal reference for quantification, and ^1^H spectra were obtained using a Bruker AM500 NMR spectrometer [50,51]. Approximately 50 μL of 10 mM DSS was used as an internal reference for quantification. The final GB concentration was determined using three independent samples.

### 4.5. Determination of Photosystem II Activity, Photosynthesis Rate, Damage to Leaf Cell Membranes, and RWC

The activity of photosystem II (*Fv/Fm*) was determined on the third leaf of each maize seedling after adaptation to dark for 30 min using a pulse-modulation chlorophyll fluorometer (FMS-2; Hansatech, Norfolk, UK) at room temperature. The net photosynthesis rate was determined on the third leaf of each maize seedling with an LI6400 portable apparatus (LI-COR, Lincoln, NE, USA) at room temperature at a PFD of 1 mmol/m^2^/s.

Leaf cell membrane damage was measured as the leakage of electrolytes from the leaf cells using a conductivity meter and calculated as described previously [52] (Premachandra et al., 1989). The equation used for estimating cell membrane damage can be represented as follows:Cell membrane damage (%) = [1 − (1 − S1/S2)/(1 − C1/C2)] × 100,
where the conductivity measurements correspond to PEG-treated leaves (S1), boiled PEG-treated leaves (S2), non-PEG-treated leaves (C1), and boiled non-PEG-treated leaves (C2). All experiments were repeated at least thrice.

### 4.6. Drought Treatment Assay of Maize Plants at the Pre-Flowering and Heading Stages

Maize seeds were grown in the soil of flowerpots (35 cm in diameter and 30 cm in height), with one plant per pot, or in wooden troughs, and watered sufficiently every 2 d. When the seedlings had grown to the 10-leaf stage, the RWC of soil was maintained at 15% for 3 weeks by decreasing the quantity of watering. Subsequently, the plants were watered sufficiently. In addition, the morphology, net photosynthetic rate, and physiological parameters were recorded.

A field experiment was conducted in the experimental field of Shandong University, Jinan, China, in a rain-exclusion shelter. Maize seeds were germinated under sufficient irrigation as required. When the plants had reached the 14-leaf stage, the RWC of soil to a depth of 40 cm was maintained at 15–19% for 2 weeks, and subsequently, the dehydration-stress-treated maize was watered sufficiently. After the ears matured, these maize plants were evaluated for their dry weight and yield traits.

### 4.7. Breeding of a Single Hybrid with Drought Tolerance

Based on the segregation ratio of the transgene, homologous transgenic lines from independent transformants were chosen, and T_2_ plants with improved drought tolerance were selected for self-pollination. T_4_ plants from homologous transgenic lines with improved drought tolerance were mated with NT commercial inbred lines, such as line 196, to produce hybrids (F_1_ seeds).

The F_1_ seeds of the hybrid lines and those derived from the NT commercial hybrid controls were sown in the plots. Each trial was divided into three blocks, and one hybrid was randomly assigned to each block. The area of each plot was 10 m^2^, with six lines per plot. Seeds of the hybrids were sown in mid-June, and observations were carried out at the five-leaf, blooming, and mature stages. The number of plants and the number and weights of ears were recorded at harvest. Thirty hybrid ears were sampled randomly, and kernel rows and kernels in each row were counted. Dry kernels were weighed, and their mean kernel weight was calculated.

### 4.8. Statistical Analyses

All data are presented as the mean ± standard deviation of at least three replicates. Statistical analysis was performed using the R software. Analysis of variance (ANOVA) and Tukey’s honestly significant difference (HSD) test were used for multiple comparisons, and Student’s *t*-test was used to compare the transgenic lines with their control counterparts. Double asterisks (**) and a single asterisk (*) denote statistical significance with *p* < 0.01 and *p* < 0.05, respectively, in Student’s *t*-test, and different letters denote statistical significance with *p* < 0.05 in ANOVA and Tukey’s HSD tests.

## 5. Conclusions

GB is a quaternary amine with zwitterionic nature, and its natural accumulation in cells is known to protect organisms against abiotic stresses such as drought and salt stress via osmoregulation or osmoprotection. However, as one of the major cereals, maize did not accumulate a significant amount of GB naturally, which could be due to the alternative splicing of the endogenous betaine aldehyde dehydrogenase coding gene. In this study, four elite inbred maize lines, DH4866, Qi319, Y478 and DH9938, were transformed with the *E. coli betA* gene, a key enzyme in the biosynthesis of GB from choline, using *Agrobacterium* to generate *betA* transgenic lines. The drought tolerance of T4 progenies derived from these transgenic plants was evaluated at germination, seedling, pre-flowering and heading stages in the lab, greenhouse and field under reduced irrigation conditions. Transgenic plants accumulated higher levels of glycine betaine and were relatively more tolerant to drought stress than NT at both the germination and early seedling stages. The grain yield of the transgenic plants was significantly higher than NT plants after drought treatment. Moreover, hybrids were obtained by crossing the drought-tolerant transgenic lines. These transgenic hybrids enhanced drought tolerance under field conditions compared to the non-transgenic (control) plants and two other commercial hybrids in China. The heterogeneous expression of *betA* can improve the viability of crops grown in soils with sufficient or insufficient water. These transgenic inbred lines and hybrids were useful in marginal and submarginal lands in semiarid and arid regions.

## Figures and Tables

**Figure 1 ijms-25-10630-f001:**
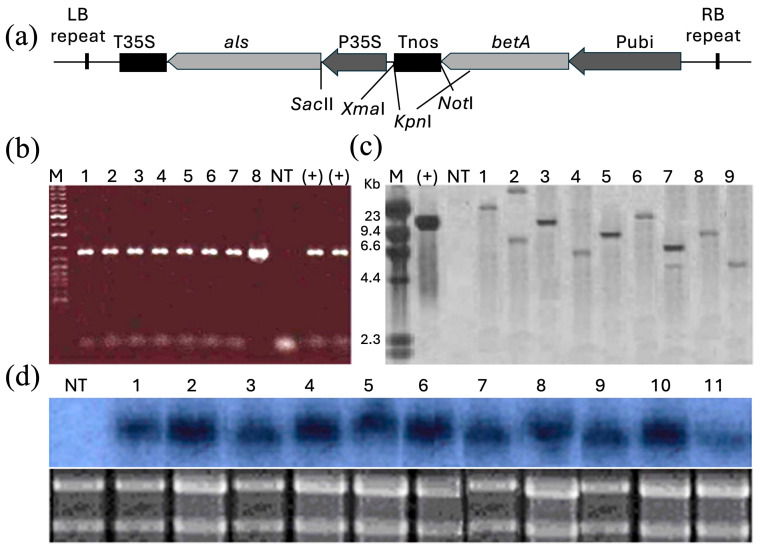
Molecular identification of maize transgenic plants. (**a**) Schematic structure of the transfer DNA region of plasmid pCU-*betA*-*als.* LB, left border; RB, right border; *betA*, CDH coding gene form *E. coli*; *als*, mutant acetolactate synthase; P35S, CMV35 promoter; Pubi, maize polyubiquitin gene promoter; T35S, CaMV poly(A) signal; Tnos, nopaline synthase terminator and poly(A) signal. *Sac*II, *Xma*I, *Kpn*I, *Not*I were unique cutters in the T-DNA region. Transgenic plants were analyzed (**b**) for *als* using PCR and (**c**,**d**) for *betA* transgene using Southern (**c**) and Northern (**d**) blotting. For Southern blotting, *Kpn*I was used to digest the genomic DNA. M, DNA ladder; 1–11, independent transgenic plants; (+), PCR amplification of plasmid pCU-*betA*-*als*; NT, non-transgenic (control).

**Figure 2 ijms-25-10630-f002:**
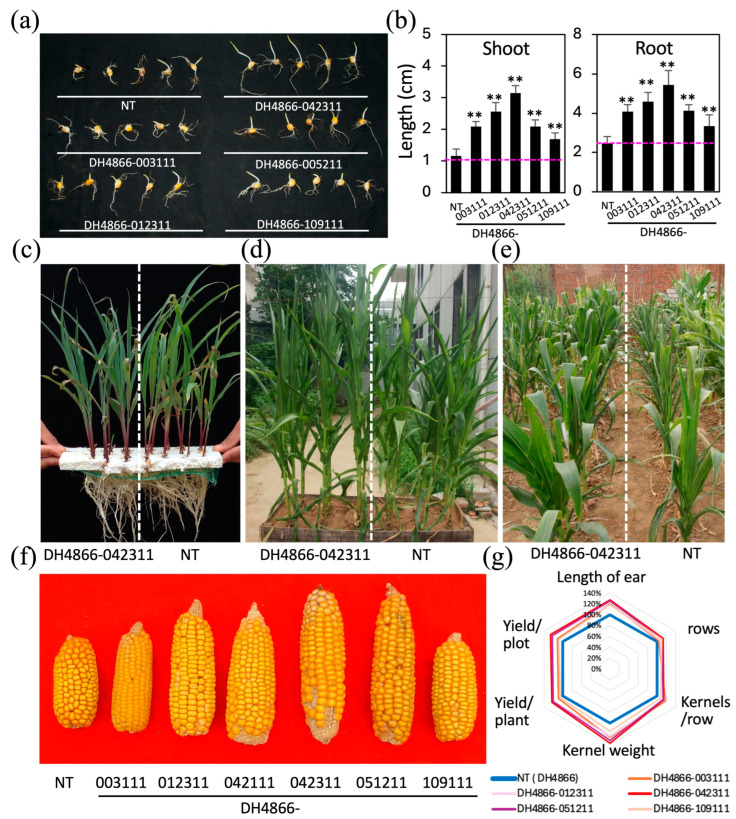
Effects of *betA* transgene on drought tolerance in maize. (**a**) Seeds of the *betA*-transgenic lines and NT plants germinated on filter paper soaked in a solution of 20% (*w*/*v*) PEG 6000 for 7 days. (**b**) Average length of the shoot and seminal roots of the seedlings in panel (**a**). Values represent means ± standard deviation from six biological replicates. Asterisks (**) indicate significant differences between the transgenic and NT lines with *p* < 0.01 based on the *t*-test. (**c**) Three-leaf stage seedlings of the *betA*-transgenic lines and the NT plants subjected to osmotic stress, i.e., treated with 14% (*w*/*v*) PEG 6000 for 7 days. (**d**,**e**) Plants derived from the *betA*-transgenic lines and the NT plants subjected to drought stress at the pre-flowering stage in wooden troughs (**d**) and at the heading stage in the field (**e**). (**f**,**g**) Ears (**f**) and yields (**g**) of the *betA*-transgenic lines and NT plants subjected to drought stress at the heading stage. The radar chart shows the changes in the agronomic traits of maize plants grown under drought-stress conditions compared with those of the NT plants grown under the same conditions. NT, non-transgenic (control). DH4866-003111, DH4866-012311, DH4866-142311, DH4866-051211, and DH4866-109111 were T_4_ plants derived from independent transgenic lines.

**Figure 3 ijms-25-10630-f003:**
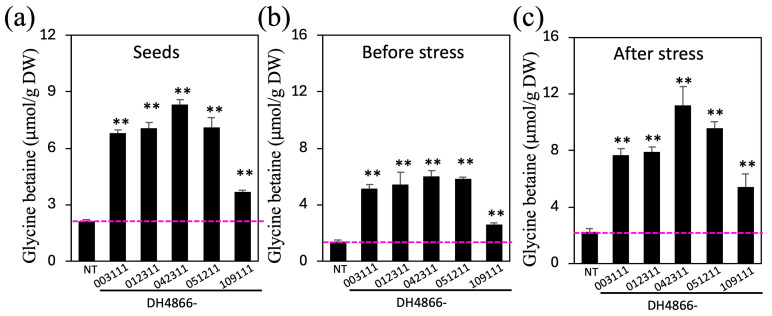
Betaine concentrations in the leaves and seeds of the NT and transgenic maize plants. (**a**–**c**) Levels of glycine betaine were determined in maize seeds (**a**) and leaves (**b**,**c**). Maize seedlings at the three-leaf stage before and after 5 days of osmotic treatment in MS salt solution containing 14% (*w*/*v*) PEG 6000 were used for the analyses, whose results are depicted in panels (**b**,**c**). Values represent means ± standard deviation from three biological replicates. Statistical analyses were performed as described in Figure 2. Asterisks (**) indicate significant differences between the transgenic and NT lines with *p* < 0.01 based on the *t*-test. NT, non-transgenic (control); DW, dry weight.

**Figure 4 ijms-25-10630-f004:**
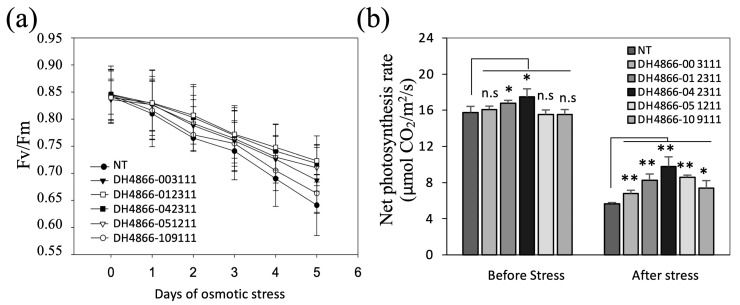
Changes in the photosystem II activity (**a**) and net photosynthesis rate (**b**) of maize seedlings under osmotic stress. (**a**) Maize seedlings at the three-leaf stage were transferred to MS salt solution containing 14% (*w*/*v*) PEG 6000 and aerated. The activity of photosystem II (*Fv/Fm*) was determined on the third leaf of each maize seedling after adaptation to dark for 30 min at 25 °C. (**b**) Net photosynthesis rate was measured on the third leaf of each maize seedling at room temperature with a PFD of 1 mmol/m^2^/s before and after 5 days of the osmotic treatment of maize seedlings at the three-leaf stage in MS salt solution containing 14% (*w*/*v*) PEG 6000. Lines and statistical analysis were the same as described in Figure 2. Asterisks (**) indicate significant differences between the transgenic and NT lines with *p* < 0.01 and asterisks (*) indicate significant differences between the transgenic and NT lines with *p* < 0.05 based on the *t*-test and n.s indicate no significant differences between the transgenic and NT lines based on the statistical analysis. NT, non-transgenic (control).

**Figure 5 ijms-25-10630-f005:**
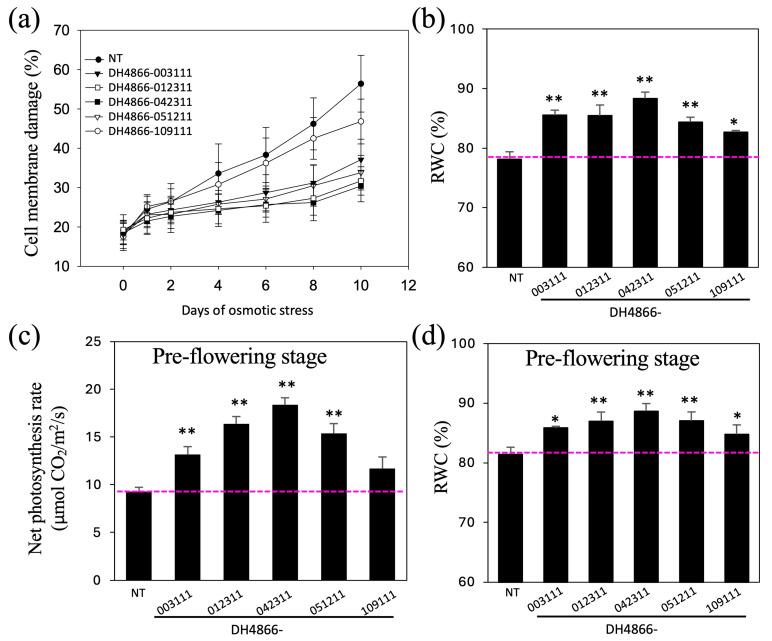
Damage to leaf cell membranes, RWC, and net photosynthesis rate of maize plants under osmotic stress. Electrolyte leakage from leaf cells (**a**) and RWC (**b**) in maize seedlings transferred to MS salt solution containing 14% (*w*/*v*) PEG 6000 and aerated at the three-leaf stage. Electrolyte leakage of leaf cells was determined every 24 h, and the RWC was recorded after 9 days of the 14% (*w*/*v*) PEG 6000 treatment. The net photosynthetic rate (**c**) and RWC (**d**) of maize plants subjected to drought stress in the field at the pre-flowering stage. Statistical analyses were performed as described in Figure 2. Asterisks (**) indicate significant differences between the transgenic and NT lines with *p* < 0.01 and asterisks (*) indicate significant differences between the transgenic and NT lines with *p* < 0.05 based on the *t*-test. RWC, relative water content; PEG, polyethylene glycol; NT, non-transgenic (control).

**Figure 6 ijms-25-10630-f006:**
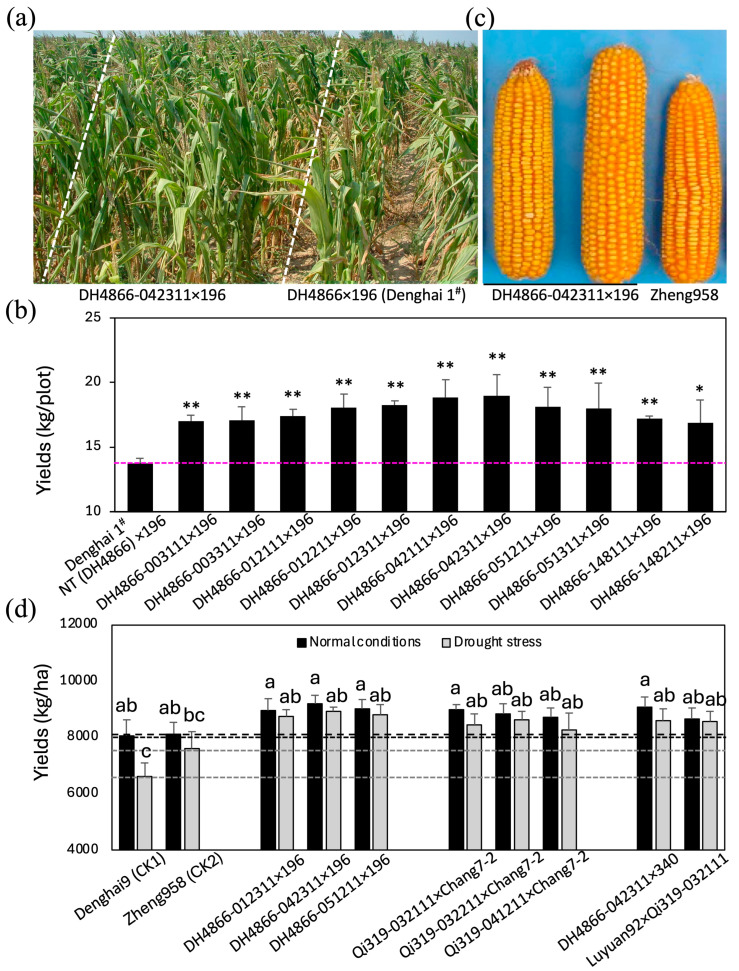
Comparison of the transgenic hybrids and NT elite hybrids. (**a**) Maize plants derived from a transgenic hybrid and NT hybrid subjected to 8 days of drought stress in the field at the heading stage. Transgenic hybrid lines DH4866-042311 × 196 and DH4866 × 196 (hybrid Denghai 1^#^) are shown. (**b**) Yields from the transgenic hybrids and a NT elite hybrid subjected to drought stress in the field at the heading stage. Hybrids were generated by crossing the male line 196 with the NT and transgenic lines derived from DH4866. Asterisks (**) indicate significant differences between the transgenic and NT lines with *p* < 0.01 and asterisks (*) indicate significant differences between the transgenic and NT lines with *p* < 0.05 based on the *t*-test. (**c**) Ears from transgenic hybrid line DH4866-042311 × 196 and Zheng 958. (**d**) Yields from the transgenic hybrids and two widely cultured commercial hybrids (Denghai 9^#^ and Zheng 958). Hybrids were generated by crossing the transgenic lines with male lines 196, 340, Chang 7-2, or Luyuan 92. Lines and statistical analysis were the same as described in Figure 2. The transgenic hybrids (F_1_) and two commercial hybrids used in China, Zheng 958 and Denghai 9^#^, were sown in plots under normal growth conditions and subjected to water deficit by stopping irrigation at the heading stage. The area of each plot was 10 m^2^, with six lines per plot. Seeds of the hybrids were sown in mid-June, and observations were carried out at the five-leaf, blooming, and mature stages of the hybrid plants. The number of plants and the number and weights of ears were recorded at harvest. Thirty ears were sampled randomly, and kernel rows and kernels in each row were counted. Dry kernels were weighed, and their mean kernel weight was calculated. Data represent mean ± standard deviation. Different letters denote statistical significance with *p* < 0.05 in ANOVA and Tukey’s HSD tests.

## Data Availability

The original contributions presented in the study are included in the article/Appendix A, further inquiries can be directed to the corresponding author.

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
