# Peer review of "Breeding Maize Hybrids with Improved Drought Tolerance Using Genetic Transformation"

_ijms, 2024, doi:10.3390/ijms251910630_

Round 1

Reviewer 1 Report

Comments and Suggestions for Authors

Dear Authors, I have reviewed the manuscript and found the following:

The topic of the manuscript is abiotic stress tolerance in genetically engineered maize. This is a very important topic, as in many countries of the world GMO maize is allowed and is an important agricultural commodity. 

Introduction: it is appropriate to start the chapter with the importance of maize in the abiotic context, and then move on to stress effects and other parts of the chapter. I also suggest writing a little more about the adverse effects and trends of climate change - I suggest adding this to the chapter. 

The figures and tables are good. 

Discussion: it would be worthwhile to include a Conclusions chapter, I feel this would be important. 

Author Response

Response to the reviewer 1’ comment:

Dear Authors, I have reviewed the manuscript and found the following:

The topic of the manuscript is abiotic stress tolerance in genetically engineered maize. This is a very important topic, as in many countries of the world GMO maize is allowed and is an important agricultural commodity. 

Thank you for your affirmation and encouragement of our work. And thanks so much for helping us a lot to improve this MS.

Q1: Introduction: it is appropriate to start the chapter with the importance of maize in the abiotic context, and then move on to stress effects and other parts of the chapter.

A1: Thanks so much for helping us a lot to improve this MS. The importance of maize in the abiotic context was aaded as suggested. See details in the introduction on Page 1, lines 33-52.

Q2: I also suggest writing a little more about the adverse effects and trends of climate change - I suggest adding this to the chapter.

A2: Thanks for your suggestion. The adverse effects and trends of climate change was aaded as suggested. See details in the introduction on Page 1,  lines 33-52..

Q3: The figures and tables are good.

A3: Thank you for your encouragement. Another reviwer mentioned that resolution of the figures were not good in the pdf version. This fixed in the revised MS.

Q4: Discussion: it would be worthwhile to include a Conclusions chapter, I feel this would be important.

Q4: Thanks for your suggestion. The Conclusions section was aaded as suggested. See details in the conclusion on Page 15, lines 576-595, acording to the formate of the journal.

Reviewer 2 Report

Comments and Suggestions for Authors

This study investigates breeding maize hybrids to improve their drought tolerance using genetic transformations. The work provides new insights compared to previous studies, which have been conducted under controlled laboratory or greenhouse conditions, not in field conditions or with maize hybrids.

The adopted method is consistent with the aims of the work and with the existing literature.

Results are very interesting, and seem to show that the introduction of glycine betaine (GB) in four maize inbred lines improves drought-stress tolerance without affecting yield potential.

The content of the manuscript is well described. Some suggestions:

·        In the ‘Introduction’ section, references are missing – Lines 70 and 74.

·        In the ‘Results’ section, the quality of figures could be improved.

·        In the ‘Discussion’ section, the advantage associated to the GB introduction by using transgenic approach could be more underlined, also mentioning relevant previous studies (for instance, https://www.ncbi.nlm.nih.gov/pmc/articles/PMC3329348/

·        https://pubmed.ncbi.nlm.nih.gov/30899269/ )

·        the ‘Conclusions’ section is absent: I suggest to add it and to stress the impact of the achieved results on future research and breeding programs.

Some minor issues:

·       An overall table of abbreviations could be added. In fact, abbreviation were explained only in figure captions

Comments on the Quality of English Language

Minor editing of English language required.

Author Response

Response to the reviewer 2’ comment:

This study investigates breeding maize hybrids to improve their drought tolerance using genetic transformations. The work provides new insights compared to previous studies, which have been conducted under controlled laboratory or greenhouse conditions, not in field conditions or with maize hybrids.

The adopted method is consistent with the aims of the work and with the existing literature.

Results are very interesting, and seem to show that the introduction of glycine betaine (GB) in four maize inbred lines improves drought-stress tolerance without affecting yield potential.

The content of the manuscript is well described. Some suggestions:

Thank you for your affirmation and encouragement of our work. And thanks so much for helping us a lot to improve this MS.

Q1: In the ‘Introduction’ section, references are missing – Lines 70 and 74.

A1: Thanks for your carefully examination. Refernces were added in those lines, refernces [1,5,35–37] and 39.

Q2: In the ‘Results’ section, the quality of figures could be improved.

A2: Thank you for your carefully examination. We tried other methods to make the figures more readable. If needed, we will ask the assistant editor for help to make it better to read.

Q3: In the ‘Discussion’ section, the advantage associated to the GB introduction by using transgenic approach could be more underlined, also mentioning relevant previous studies (for instance, https://www.ncbi.nlm.nih.gov/pmc/articles/PMC3329348/ https://pubmed.ncbi.nlm.nih.gov/30899269/ )

Q3: Thanks for your suggestion. The the advantage associated to the GB introduction by using transgenic approach was aaded as suggested. And the paper from  Annunziata et al., 2019 and Girl 2011 were added and disscussed in the revised MS. See details in the disscussion on Page 12, lines 450-462.

Q4: the ‘Conclusions’ section is absent: I suggest to add it and to stress the impact of the achieved results on future research and breeding programs.

Q4: Thanks for your suggestion. The Conclusions section was aaded as suggested. See details in . See details in the conclusion on Page 15, lines 576-595.

Q5: Some minor issues: An overall table of abbreviations could be added. In fact, abbreviation were explained only in figure captions.

A5: Thanks so much for helping us to improve this MS. An abbreviations list was added in the revised MS, P15, lines 597-600.